# Risk Association of Liver Cancer and Hepatitis B with Tree Ensemble and Lifestyle Features

**DOI:** 10.3390/ijerph192215171

**Published:** 2022-11-17

**Authors:** Eunji Koh, Younghoon Kim

**Affiliations:** 1School of Industrial and Management Engineering, Korea University, 145 Anamro, Seongbuk-gu, Seoul 02841, Republic of Korea; 2Department of Industrial and Management Systems Engineering, Kyung Hee University, 1732, Deogyeong-daero, Giheung-gu, Yongin-si 17104, Republic of Korea

**Keywords:** random forest, public health data, risk association model, liver cancer, hepatitis B, lifestyle features

## Abstract

The second-largest cause of death by cancer in Korea is liver cancer, which leads to acute morbidity and mortality. Hepatitis B is the most common cause of liver cancer. About 70% of liver cancer patients suffer from hepatitis B. Early risk association of liver cancer and hepatitis B can help prevent fatal conditions. We propose a risk association method for liver cancer and hepatitis B with only lifestyle features. The diagnostic features were excluded to reduce the cost of gathering medical data. The data source is the Korea National Health and Nutrition Examination Survey (KNHANES) from 2007 to 2019. We use 3872 and 4640 subjects for liver cancer and hepatitis B model, respectively. Random forest is employed to determine functional relationships between liver diseases and lifestyle features. The performance of our proposed method was compared with six machine learning methods. The results showed the proposed method outperformed the other methods in the area under the receiver operator characteristic curve of 0.8367. The promising results confirm the superior performance of the proposed method and show that the proposed method with only lifestyle features provides significant advantages, potentially reducing the cost of detecting patients who require liver health care in advance.

## 1. Introduction

The liver is the largest internal organ responsible for crucial body functions, such as blood coagulation, protein production, and glycogen synthesis [1,2,3,4]. Despite the importance of the organ, liver cancer is the second-largest cause of death in Korea [5]. The leading cause of liver cancer is a liver disease induced by obesity and alcohol, and hepatitis B [6]. Particularly, 70% of liver cancer patients are afflicted with hepatitis B. Hepatitis B is a disease caused by the immune response of the body to the hepatitis B virus (HBV) or contact with contaminated inanimate objects and blood [7]. It is challenging to treat effectively in the last stage of liver cancer, so that the best way to lower liver cancer mortality is to prevent the transition from hepatitis B or detect liver cancer in the early stage [8].

Disease risk association has taken a significant role in the medical research field [9]. The main objective of disease risk association is detecting diseases in the early stage to prevent leading to fatal conditions of patients [10]. We can obtain a variety of medical data and apply modern machine learning methods to predict disease risk accurately [11]. The machine learning methods have the potentials to find significant relationships between patients’ personalized data and medical diagnosis results [12].

Results of previous studies on the risk association for liver cancer have been introduced as follows: Wu et al. proposed a risk-classifier of liver cancer based on a support vector machine (SVM) with 22 biomarkers such as alpha-fetoprotein [13]. Chen et al. proposed to detect the risk of liver cancer by using clinic data such as alkaline phosphatase and aspartate transaminase and compared performances of logistic regression, decision tree, and SVM. Previous studies predict the risk of liver cancer by using clinic data and machine learning methods [14].

Other studies have attempted to predict risk for various diseases. Ward et al. proposed atherosclerotic cardiovascular disease risk association based on machine learning method with clinical data including cholesterol levels and blood pressure [15]. Perveen et al. attempted to predict the risk of non-alcoholic fatty liver disease with a decision tree and medical diagnosis results such as high-density lipoprotein and triglycerides [16]. Dimopoulos et al. proposed machine learning methods using both clinic and lifestyle features to predict cardiovascular risk [17].

The previous studies have shown promising results in risk association of liver and various diseases; however, they employ clinical diagnosis results. The clinical data enhances the risk association model’s performance with personalized information. The approaches have an explicit limitation to achieving the objective of early screening of patients. Leveraging the data from various treatments in hospitals takes considerable time, and costs [18,19]. In contrast, lifestyle features are relatively cost-effective and publicly available.

In this study, we propose a random forest-based method to predict the risk of liver cancer and hepatitis B only using lifestyle features. In consideration of related works which imply significant correlations between liver disease and lifestyle features [20,21], we assume it is possible to predict disease risk reliably with lifestyle features only. We introduce models that can predict liver cancer and hepatitis B by exploiting the same lifestyle features. The model predicts the risk cost-effectively for high-risk people with liver cancer and hepatitis B. Furthermore, it has a broader impact on managing liver health regardless of the accessibility to medical institutions.

The remainder of this paper is organized as follows. Section 2 details data and prediction methods, describing the input data, modeling, and sensitivity analysis. Section 3 presents a performance evaluation for risk association of hepatitis B and liver cancer. Section 4 discusses comparative study and sensitivity analysis results. Finally, Section 5 offers our conclusive comments.

## 2. Material and Methods

### 2.1. Data Source

We use data from the Korea National Health and Nutrition Examination Survey (KNHANES) [22] which is publicly available and consists of health status, examination, and nutrition survey. Data collection is conducted annually to evaluate Koreans’ health and nutritional status and monitor trends in health risk factors and the prevalence of major chronic diseases according to Korean Article 16 of the National Health Promotion Act [22]. Therefore, the target population of KNHANES comprises Korean citizens residing in Korea, and representative samples for the survey are newly extracted every year to obtain generalizable survey data according to the sampling plan. The sampling plan follows a multi-stage clustered probability design. We use KNHANES data gathered over 13 years, from 2007 to 2019, as cross-sectional data to develop risk association models. It has 510,747 cases, and adult data were extracted. We selected 13 features closely related to the lifestyle. The list of desired features is presented in Table 1. We use the KNHANES dataset to develop models which detect potential patients who require liver health care in advance.

### 2.2. Data Preprocessing

We extracted liver cancer and hepatitis B samples to train risk association models. Figure 1 shows the data preprocessing flow configuring the liver cancer and hepatitis B dataset. The hepatitis B dataset consists of 856 subjects. The 58 subjects in the liver cancer dataset have experienced liver cancer. All cases with hepatitis B and liver cancer are provided by the national health and nutrition survey. Each dataset was divided into train and test sets. Train sets account for 70% of entire data sets, and test sets for 30%. Additionally, we construct train sets by randomly sampling subjects with and without a liver disease diagnosis in equal proportion to handle the class imbalance. By sampling subjects with and without a liver disease diagnosis to construct train sets in equal proportion, we improve the models to learn both majority class and minority class in equal weight. The train set consists of 82 and 1198 subjects sampled from liver cancer and hepatitis B datasets, respectively. Half of the train set is the liver disease cases, and the other half is cases without liver disease.

We imputed missing values in the dataset with mice [23], simple, and K-nearest neighbor (K-NN) imputation [24,25]. The final imputation method is determined by a comparative study. We present the imputation study result in Section 3.

### 2.3. Risk Association Model

We adopt the random forest for the risk association method. Random forest is representative ensemble method based on decision tree and have shown promising results in disease risk association [26,27]. We validate the random forest with grid search-based cross-validation (GridSearchCV) [28] to determine optimal hyperparameters. GridSearchCV validates all combinations of hyperparameters in terms of validation accuracy. Table 2 and Table 3 show the result of hyperparameter tuning. After tuning the hyperparameters, we compared the performances in terms of accuracy and an area under receiver operator characteristic curve (AUC) [29].

### 2.4. Performance Evaluation and Model Interpretation

We examine the validity of proposed method by comparing with logistic regression [30], decision tree [31], artificial neural networks(ANN) [32], extreme gradient boosting(XGboost) [33], light gradient boosting machine(lightGBM) [34], and SVM [35]. The metrics for evaluation are AUC and accuracy. The AUC is the representative value-based metric for risk association models by quantitatively measuring the entire two-dimensional area under the receiver operating characteristic (ROC) curve [36,37,38]. It measures how much the model is capable of distinguishing between classes. The higher the AUC, the better the model is in distinguishing between classes. In the medical field, previous studies generally used AUC as an evaluation index for disease risk association [39,40,41]. Accuracy is an index of the proportion of correctly predicted cases among all cases. We averaged the validation results repeated 30 times and calculated the standard deviation.

We conducted a sensitivity analysis after training prediction models to interpret the training results [42]. Random forest is usually more accurate than other models, but it has the disadvantage of being difficult to interpret. Even the decision tree, the base model for random forest, is interpretable, ensembe makes the random forest lose the interpretability. To address the issue, we performed a sensitivity analysis in two methods.

As the first method, we performed a sensitivity analysis by randomly permuting one feature in the dataset and then retraining with the same model. We can analyze the result whether the corresponding feature significantly affects the performance of disease risk association by observing significant drop in the model’s performance after the re-fitting. We identified the performance drop with AUC. The second method is a Shapley additive explanation (SHAP) [43]. SHAP captures the influence of each feature on the model’s performance through the Shapley value, the average marginal contribution of a feature over all possible coalitions of features. Therefore, we can identify significant features having a large Shapley value.

## 3. Case Study Results

### 3.1. Results of Imputations

We compare the three imputation methods: simple imputation, K-NN imputation, and mice imputation. Table 4 and Table 5 illustrate the comparison results over imputation methods by AUC. In the risk association model for liver cancer, simple imputation achieves the highest AUC of 0.8317. The overall results imply that the simple imputation is the best for imputing missing values in our dataset in all models as well as random forest. In the risk association model for hepatitis B, mice imputation achieves the highest AUC of 0.8632. Mice imputation shows better performance than simple imputation. We subsequently use the simple imputation and mice imputation for liver cancer and hepatitis B risk association, respectively.

### 3.2. Performance Evaluation

Table 6 shows evaluation results over comparative methods. Both risk association models with random forest for hepatitis B and liver cancer performed well with AUC values of 0.8367 and 0.88083. Values of accuracy in the test set were 0.9245 for the hepatitis B risk association model and 0.8190 for the risk association model for liver cancer. The overall results demonstrate that the proposed method outperformed other comparative methods.

In addition, to demonstrate the superiority of the proposed method over comparative methods, we conducted Wilcoxon signed-rank test on the random forest and comparative methods with rank statistics. In each validation step, we calculate the rank of the methods, and after validation, the number of rank results for each method is equal to 30. The 30 rank results are statistically compared with Wilcoxon signed-rank test. Table 7 shows the Wilcoxon signed-rank test results between random forest and other comparative methods. The result demonstrates the statistical significance between the random forest and others. In Table 7, all p-values are less than 0.05. The result demonstrates that the performance gap between random forest and others is statistically significant.

### 3.3. Sensitivity Analysis

We conducted a sensitivity analysis to identify which features significantly improve random forest’s performance using two approaches. The first method randomly permutates a specific feature and then retrains the random forest model with the randomly permuted feature. Sensitivity analysis also used AUC to evaluate the performance. We average the AUC values obtained from 30 retrained models for accurate comparison and calculate the standard deviation. The second method uses SHAP to evaluate the influence of each feature on random forest model learning. Similarly, we average the each feature’s SHAP values obtained from test dataset for an accurate comparison.

We established the hypothesis based on well-known prior studies [44,45,46] on the correlation between liver health and lifestyle features. The established hypotheses are: first, high body mass index (BMI) is closely related to the risk of hepatitis B and liver cancer because it can cause various adult diseases. BMI is the weight (kg) divided by the height (m) square, commonly used to measure a person’s health condition [44]. Second, hepatitis B is generally known as a high-risk factor for liver cancer [45]. It is assumed that hepatitis B is a noteworthy feature in the risk model of liver cancer. Finally, drinking is closely related to liver health [46]. Therefore, we assume that the drinking age will significantly impact hepatitis B and liver cancer risk.

We interpret the sensitivity analysis result according to the above hypothesis. Table 8 shows no significant difference between the sensitivity analysis results of the BMI, and the existing model’s performance. Interestingly, average weekend sleep time per day is the noteworthy feature that causes the most considerable performance degradation in the model. The difference of mean AUC between original model and permutated model by the average weekend sleep time per day is 0.0311. Considering the confidence intervals, the difference is significant.

In Table 9, we present the sensitivity analysis results of the liver cancer risk association. The drinking age, average weekend sleep time per day, hepatitis B, and weight affect the performance of the prediction model. As a result of hypothesis testing, the hepatitis B and BMI have relatively large effects compared to other features, but the drinking age has no significant impact on performance. The hepatitis B causes the most critical performance degradation of mean AUC 0.0432.

As shown in Table 10, we present the SHAP results of the hepatitis B risk association. The average weekend sleep time per day, weekday sleep time per day, and total monthly household income have relatively larger influence on the prediction model than other features. As a result of hypothesis testing, the drinking age has no significant impact on performance. The SHAP value of the drinking age is 0.0190. Table 11 shows the age and hepatitis B have relatively significant influence on prediction model’s performance. However, the drinking age is the feature that causes the no considerable influence on performance.

We interpret the models in sensitivity analysis and SHAP. The overall results are meaningful because it can explain features that have a great effect on the prediction models. Table 12 shows the top 5 features of sensitivity analysis and SHAP about hepatitis B risk association models. There are four in common features (average weekend sleep time per day, average weekday sleep time per day, average weekly working hours, and age) that have relatively significant effects on model’s performance. Table 13 demonstrates that hepatitis B, age, and total monthly household income have relatively large impact on the liver cancer risk association model.

Table 14 and Table 15 present the example of SHAP analysis for each observation and show how to interpret significant features involved in each observation. The SHAP results of each observation with liver and non-liver cancer are shown in Table 14. Age and hepatitis B are common features that significantly affect model performance. However, depending on the personal history of liver cancer for each observation, there is a difference in how lifestyle features affect the risk association of liver cancer with the model. For instance, the SHAP value of hepatitis B in the observation with liver cancer is 0.1450. It shows that hepatitis B leads the model to predict the observation having a high risk of liver cancer. However, the SHAP value of hepatitis B in the observation without liver cancer is −0.0639. It presents that non-hepatitis B forces the model to predict that the observation has a low risk of liver cancer.

Table 15 shows the SHAP results of each observation with hepatitis B or non-hepatitis B. Three common features (average weekday sleep time per day, average weekend sleep time per day, and average weekly working hours) significantly affect the model’s performance. Observation with hepatitis B has shorter weekday and weekend sleep time per day than the observation without hepatitis B. The SHAP values of average weekday sleep time per day and average weekend sleep time per day in observation with hepatitis B are 0.1811 and 0.2252, respectively. Conversely, in the observation without hepatitis B, the SHAP value of average weekday sleep time per day is −0.1255, and the average weekend sleep time per day is −0.1111. Therefore, the results imply a positive correlation between average sleep time per day and the risk of hepatitis B.

### 3.4. Ablation Studies

We conducted ablation studies to evaluate models’ performance in various experiment settings. We present the changes of model performance according to the number of training datasets in Figure 2 and Figure 3. As can be seen in Figure 2 and Figure 3, random forest outperformed other comparative methods in the all training dataset ratio settings. Note that in Figure 2, the result for random forest and the lightGBM is relatively close. To objectively examine the performance, we refer Table 16 and Table 17, where the AUC results are averaged over 30 validation runs.

The results shown in Table 16 and Table 17 illustrate random forest results over comparative methods. Both risk association models with random forest for hepatitis B and liver cancer performed better than other methods in the all training dataset ratio. AUC values in the test set were 0.8288 for the hepatitis B risk association model and 0.7963 for the risk association model for liver cancer when they used 10% of the training dataset. The results confirm that the proposed approach is robust to real-world small data problems.

In addition, we conducted a Wilcoxon signed-rank test by setting the Type I error α to 0.05 to evaluate the robustness of random forest to small training data. Table 18 shows the Wilcoxon signed-rank test results between random forest and other comparative methods when they used 10% of the training dataset. In Table 18, there was a clear performance difference between the random forest and the other comparative methods in liver cancer risk association. Moreover, the P-values of the comparative methods are less than 0.05 in the hepatitis B risk association. We verify that random forest had robustness to real-world small data problems.

We evaluate the performance of multi-disease risk association models and comparative models. Multi-disease risk association models predict both liver cancer and hepatitis B risk simultaneously. Table 19 illustrates the comparison results over seven models by AUC. Note that the test results of hepatitis B do not exceed 0.61. Even though we optimize hyperparameters of each model, they have no remarkable performance improvements. An AUC value of hepatitis B from a random forest is only about 0.5350. The test results of hepatitis B shown in Table 19 validate that it is not comparable to our method to predict hepatitis B risk. Consequently, we believe that building new models for each liver disease is desirable.

## 4. Discussion

Risk association models generally use clinical data requiring costly and time-consuming procedures [39,40,47]. However, the lifestyle features are much more accessible and cost-efficient [48]. We selected lifestyle features related to liver disease by referring to previous studies. All citizens in Korea subscribe to health insurance through the national insurance system [49]. The types of national insurance include local subscribers, work subscribers, and medical benefits. According to a preliminary study, the status of medical service usage, such as vaccination, is highly correlated with the type of national insurance subscription [50]. Based on the result, we assume that individuals’ medical life, such as vaccination, can be identified through the type of national insurance subscription and private insurance subscription. Therefore, we included a feature for identifying the subscription status of health insurance.

Additionally, referring to the previous research [51], the transmission of the hepatitis B virus occurs in contact with body fluids or contaminated inanimate organisms. Sufficient protein intake and rest can help recover from hepatitis B [52]. Thus, we included dietary life, sleep time, and working environment features.

The comparison study results verify that the random forest is superior to comparative methods (logistic regression, decision tree, ANN, XGboost, lightGBM, and SVM) in the risk association of liver disease. The logistic regression and decision tree lag behind the random forest because the methods are appropriate for linear or simple data distribution. The logistic regression is based on the linear transformation of input features and nonlinear activation. The capability of representing complex data distribution is generally weaker than the random forest. The decision tree divides data space with a vertical line and has limited capability in representing nonlinear data distributions [53]. Although the ANN can model the nonlinear relationships between input and outputs, it is not enough to perform better than a random forest. Variance reduction of ensemble significantly enhances the prediction performance compared to single classifiers. XGboost and LightGBM are boosting models showing comparable performance against the random forest. However, in the liver disease data, the variance reduction effect of the bagging procedure dominates the boosting effects.

Various studies have used the random forest for disease risk association in the medical field. Xu et al. [26] built a model to predict type II diabetes risk using clinical data, and it had an accuracy of 0.8413. Kabiraj et al. [54] established a risk association model for breast cancer. They trained the random forest and the XGboost classifier. The random forest achieves the best performance with an accuracy of 74.73%. The model was trained with clinical data such as start tumor size, end tumor size, irradiate, and menopause. Hashem et al. [55] developed a model with the accuracy of 0.7381 to predict hepatocellular carcinoma, which is malignant liver cancer. They adopted the random forest and calibrated the model with clinical characteristics, including hemochromatosis, arterial hypertension, and chronic renal insufficiency. Although our proposed method is trained with lifestyle features, it shows comparable or superior performance in terms of AUC and accuracy among related works.

## 5. Conclusions

We proposed a risk association method for liver cancer and hepatitis B with random forest. The proposed method showed superior performance over comparative methods. In particular, the results of the risk association of hepatitis B verified that the proposed method is more robust than different approaches. The sensitivity analysis demonstrated that some lifestyle features strongly correlate with hepatitis B. The results can potentially reduce the cost of detecting potential patients who require liver health care in advance.

## Figures and Tables

**Figure 1 ijerph-19-15171-f001:**
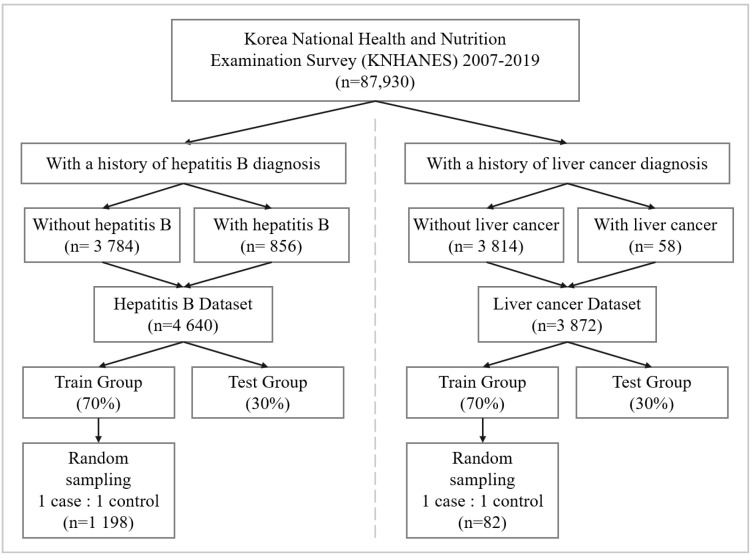
Workflow of producing data sets for the prediction modeling. The liver cancer and hepatitis B datasets consist of 3872 and 4640 subjects, respectively. The datasets are divided into 70% training data and 30% testing data.

**Figure 2 ijerph-19-15171-f002:**
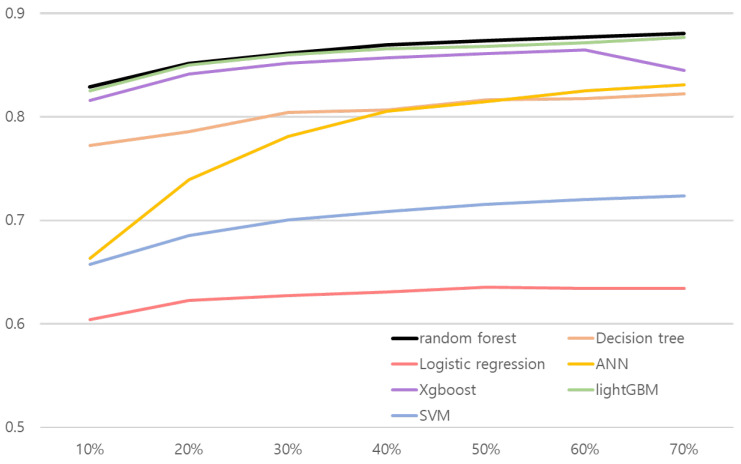
Line plot of hepatitis B risk association model’s performance according to the number of train dataset. *x* and *y* axes represent the proportion of training data and AUC, respectively.

**Figure 3 ijerph-19-15171-f003:**
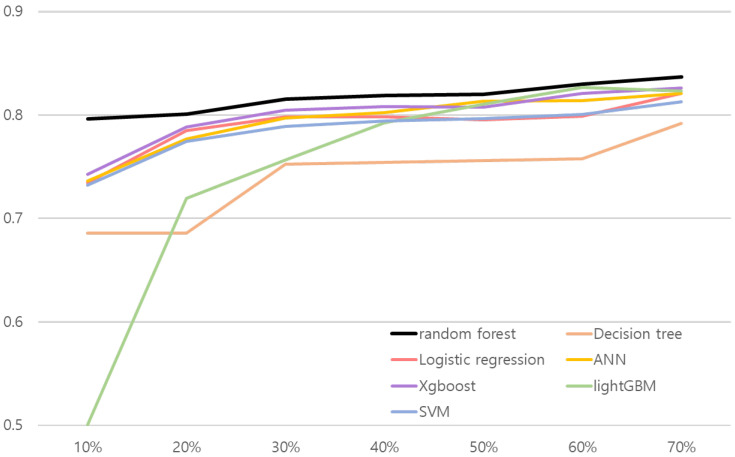
Line plot of liver cancer risk association model’s performance according to the number of train dataset. *x* and *y* axes represent the proportion of training data and AUC, respectively.

**Table 1 ijerph-19-15171-t001:** Input features for the proposed methods. There are nine continuous features and three categorical features.

Type	Feature
Continuous	Age
	Height
	Weight
	BMI
	Drinking age
	Average weekday sleep time per day
	Average weekend sleep time per day
	Total monthly household income
	Average weekly working hours
Categorical	National health insurance type
	Private health insurance status
	Hepatitis B
	Sex

**Table 2 ijerph-19-15171-t002:** List of optimized hyperparameters for the random forest.

Model	Parameter	Liver Cancer	Hepatitis B
Random forest	N estimators	300	100
	Max depth	30	10
	Criterion	entropy	entropy
	Min samples split	5	10
	Min sample leaf	1	1
	Bootstrap	True	True
	Warm strat	True	False
	Max features	auto	auto

**Table 3 ijerph-19-15171-t003:** List of optimized hyperparameters for the comparative models.

Model	Parameter	Liver Cancer	Hepatitis B
Decision tree	Max depth	10	10
	Criterion	entropy	entropy
	Min samples split	2	5
	Min sample leaf	1	1
Logistic regression	Class weight	none	balanced
	Max iter	50	50
ANN	Slover	sgd	adam
	Activation	tanh	tanh
	Alpha	0.0001	0001
	Learning rate	constant	constant
	Hidden layer size	(100, 50)	(50, 25)
XGboost	Eta	0.1	0.3
	Min child weight	0	5
	Max depth	6	6
	Gamma	0	5
	Colsample bytree	0.5	0.6
	N estimators	100	100
LightGBM	Num leaves	31	5
	Max depth	−1	−1
	N estimator	1	100
	Learning rate	0.05	0.1
	Min data in leaf	10	20
	Num iteration	100	100
	Colsample bytree	0.6	1.0
	Boosting	dart	gbdt
SVM	Kernel	sigmoid	rbf
	Degree	1	1
	Gamma	auto	scale

**Table 4 ijerph-19-15171-t004:** Average AUC values for each combination of the liver cancer risk association model and missing value imputer. Standard deviation is calculated in the parentheses.

	Risk Association Model for Liver Cancer		
	Simple Imputation	K-NN Imputation	Mice Imputation
Random forest	0.8317 (0.0345)	0.8062 (0.0501)	0.8248 (0.0276)
Decision tree	0.7671 (0.0531)	0.7546 (0.0569)	0.7710 (0.0543)
Logistic regression	0.8211 (0.0447)	0.8186 (0.0469)	0.8199 (0.0539)
ANN	0.8129 (0.0431)	0.8066 (0.0463)	0.8129 (0.0393)
XGboost	0.8153 (0.0433)	0.7788 (0.0587)	0.8154 (0.0411)
lightGBM	0.7992 (0.0464)	0.7636 (0.0419)	0.7925 (0.0447)
SVM	0.8098 (0.0555)	0.8025 (0.0581)	0.8055 (0.0470)

**Table 5 ijerph-19-15171-t005:** Average AUC values for each combination of the hepatitis B prediction model and missing value imputer. Standard deviation is calculated in the parentheses.

	Risk Association Model for Hepatitis B		
	Simple Imputation	K-NN Imputation	Mice Imputation
Random forest	0.8567 (0.0124)	0.7050 (0.0147)	0.8632 (0.0113)
Decision tree	0.7818 (0.0132)	0.6295 (0.0178)	0.7947 (0.0157)
Logistic regression	0.6384 (0.0144)	0.6313 (0.0147)	0.6338 (0.0137)
ANN	0.8015 (0.0121)	0.6548 (0.0120)	0.7990 (0.0150)
XGboost	0.8469 (0.0108)	0.7505 (0.0119)	0.8611 (0.0094)
lightGBM	0.8534 (0.0115)	0.7520 (0.0126)	0.8636 (0.0098)
SVM	0.7401 (0.0149)	0.6560 (0.0167)	0.7235 (0.0127)

**Table 6 ijerph-19-15171-t006:** Performances of risk association models and comparative models. Standard deviation is calculated in the parentheses.

	Liver Cancer Risk Association		Hepatitis B Risk Association	
	AUC	Accuracy	AUC	Accuracy
Random forest	0.8367 (0.0357)	0.8190 (0.0384)	0.8803 (0.0030)	0.9245 (0.0060)
Decision tree	0.7921 (0.0568)	0.7807 (0.0481)	0.8224 (0.0206)	0.8423 (0.0305)
Logistic regression	0.8211 (0.0447)	0.8302 (0.0289)	0.6341 (0.0136)	0.6541 (0.0159)
ANN	0.8206 (0.0526)	0.8312 (0.0311)	0.8306 (0.0142)	0.8466 (0.0111)
XGboost	0.8260 (0.0392)	0.8075 (0.0409)	0.8778 (0.0103)	0.9309 (0.0057)
LightGBM	0.8234 (0.0474)	0.8118 (0.0485)	0.8764 (0.0117)	0.9299 (0.0068)
SVM	0.8130 (0.0507)	0.8408 (0.0351)	0.7235 (0.0127)	0.6919 (0.0137)

**Table 7 ijerph-19-15171-t007:** Results of Wilcoxon signed-rank test between random forest and comparative models.

	*p*-Value	
	Liver Cancer Risk Association	Hepatitis B Risk Association
Decision tree	0.0000	0.0000
Logistic regression	0.0210	0.0000
ANN	0.0336	0.0000
XGboost	0.0074	0.0000
LightGBM	0.0408	0.0022
SVM	0.0042	0.0000

**Table 8 ijerph-19-15171-t008:** Sensitivity analysis results of hepatitis B risk association model. Standard deviation is presented in parentheses.

Permutated Feature	Mean of AUC
-	0.8803 (0.0030)
Age	0.8750 (0.0069)
Sex	0.8799 (0.0050)
Height	0.8795 (0.0051)
Weight	0.8803 (0.0051)
BMI	0.8788 (0.0062)
Total monthly household income	0.8773 (0.0070)
Average weekly working hours	0.8664 (0.0060)
Average weekday sleep time per day	0.8516 (0.0090)
Average weekend sleep time per day	0.8492 (0.0087)
Drinking age	0.8710 (0.0154)
Private health insurance status	0.8767 (0.0064)
National health insurance type	0.8790 (0.0054)

**Table 9 ijerph-19-15171-t009:** Sensitivity analysis result of the risk association model for liver cancer. Standard deviation is presented in parentheses.

Feature	Mean of AUC
-	0.8367 (0.0357)
Age	0.8126 (0.0483)
Sex	0.8216 (0.0368)
Height	0.8327 (0.0376)
Weight	0.8255 (0.0429)
BMI	0.8253 (0.0424)
Total monthly household income	0.8229 (0.0444)
Average weekly working hours	0.8312 (0.0352)
Average weekday sleep time per day	0.8288 (0.0429)
Average weekend sleep time per day	0.8301 (0.0428)
Drinking age	0.8307 (0.0378)
Private health insurance status	0.8327 (0.0374)
Hepatitis B	0.7935 (0.0379)
National health insurance type	0.8344 (0.0374)

**Table 10 ijerph-19-15171-t010:** SHAP values of hepatitis B risk association model.

Feature	SHAP Value
Age	0.0476
Sex	0.0093
Height	0.0089
Weight	0.0122
BMI	0.0116
Total monthly household income	0.0482
Average weekly working hours	0.0349
Average weekday sleep time per day	0.1240
Average weekend sleep time per day	0.1175
Drinking age	0.0190
Private health insurance status	0.0290
National health insurance type	0.0019

**Table 11 ijerph-19-15171-t011:** SHAP values of the risk association model for liver cancer.

Feature	SHAP Value
Age	0.0940
Sex	0.0090
Height	0.0159
Weight	0.0219
BMI	0.0147
Total monthly household income	0.0421
Average weekly working hours	0.0499
Average weekday sleep time per day	0.0327
Average weekend sleep time per day	0.0748
Drinking age	0.0200
Private health insurance status	0.0229
Hepatitis B	0.0776
National health insurance type	0.0021

**Table 12 ijerph-19-15171-t012:** The top 5 significant features of the sensitivity analysis for hepatitis B risk association model.

Sensitivity Analysis	SHAP
Average weekend sleep time per day	Average weekday sleep time per day
Average weekday sleep time per day	Average weekend sleep time per day
Average weekly working hours	Total monthly household income
Drinking age	Age
Age	Average weekly working hours

**Table 13 ijerph-19-15171-t013:** The top 5 significant features of the sensitivity analysis for liver cancer risk association model.

Sensitivity Analysis	SHAP
Hepatitis B	Age
Age	Hepatitis B
Sex	Average weekend sleep time per day
Total monthly household income	Average weekly working hours
BMI	Total monthly household income

**Table 14 ijerph-19-15171-t014:** Example of SHAP analysis for observations of the liver cancer risk association model.

	Liver Cancer		Non-Liver Cancer	
Feature	Value	SHAP Value	Value	SHAP Value
Age	72	0.1451	53	−0.0269
Sex	Man	0.0087	Man	0.0094
Height	169.7	0.0180	168.3	−0.0088
Weight	71	0.0193	70.4	0.0072
BMI	24.7	0.0042	24.9	−0.0104
Total monthly household income	250	0.0172	390	−0.0281
Average weekly working hours	21	0.0206	40	0.0286
Average weekday sleep time per day	420	0.0319	440	0.0134
Average weekend sleep time per day	480	0.0590	440	−0.0268
Drinking age	26	0.0217	21	0.0153
Private health insurance status	Subscriber	−0.0096	Subscriber	−0.0326
Hepatitis B	Yes	0.1450	No	−0.0639
National health insurance type	Employee	−0.0015	Employee	−0.0073

**Table 15 ijerph-19-15171-t015:** Example of SHAP analysis for observations of the hepatitis B risk association model.

	Hepatitis B		Non-Hepatitis B	
Feature	Value	SHAP Value	Value	SHAP Value
Age	52	0.0295	57	0.0093
Sex	Man	0.0063	Woman	−0.0121
Height	164.1	0.0063	155.9	−0.0204
Weight	56.7	0.0138	58.6	−0.0038
BMI	21.1	0.0036	24.1	−0.0027
Total monthly household income	126.7	0.0846	166.7	0.0159
Average weekly working hours	20	−0.0770	60	−0.0315
Average weekday sleep time per day	426.5	0.1811	540	−0.1255
Average weekend sleep time per day	460.7	0.2252	540	−0.1111
Drinking age	16	−0.0034	45	0.0186
Private health insurance status	Subscriber	−0.0070	Subscriber	−0.0246
National health insurance type	Non-employee	0.0011	Non-employee	−0.0058

**Table 16 ijerph-19-15171-t016:** Changes of hepatitis B risk association model’s performance according to the number of train dataset.

Train Dataset Ratio	10%	20%	30%	40%	50%	60%	70% (Default)
Random forest	0.8288 (0.0025)	0.8513 (0.0043)	0.8614 (0.0023)	0.8691 (0.0038)	0.8732 (0.0044)	0.8771 (0.0022)	0.8803 (0.0030)
Decision tree	0.7723 (0.0269)	0.7854 (0.0241)	0.8039 (0.0133)	0.8066 (0.0130)	0.8161 (0.0147)	0.8172 (0.0135)	0.8224 (0.0206)
Logistic regression	0.6040 (0.0173)	0.6223 (0.0101)	0.6274 (0.0119)	0.6309 (0.0086)	0.6351 (0.0100)	0.6343 (0.0121)	0.6341 (0.0136)
ANN	0.6633 (0.0273)	0.7390 (0.0253)	0.7808 (0.0161)	0.8055 (0.0105)	0.8148 (0.0084)	0.8250 (0.0091)	0.8306 (0.0143)
XGboost	0.8157 (0.0108)	0.8415 (0.0075)	0.8515 (0.0066)	0.8567 (0.0066)	0.8610 (0.0074)	0.8647 (0.0099)	0.8778 (0.0103)
Light GBM	0.8253 (0.0167)	0.8497 (0.0098)	0.8597 (0.0086)	0.8659 (0.0079)	0.8678 (0.0071)	0.8712 (0.0107)	0.8764 (0.0117)
SVM	0.6576 (0.0165)	0.6855 (0.0119)	0.7001 (0.0092)	0.7083 (0.0106)	0.7156 (0.0129)	0.7202 (0.0119)	0.7235 (0.0127)

**Table 17 ijerph-19-15171-t017:** Changes of liver cancer risk association model’s performance according to the number of train dataset.

Train Dataset Ratio	10%	20%	30%	40%	50%	60%	70% (Default)
Random forest	0.7963 (0.0102)	0.8011 (0.0371)	0.8151 (0.0276)	0.8191 (0.0321)	0.8199 (0.0326)	0.8296 (0.0418)	0.8367 (0.0357)
Decision tree	0.6855 (0.0941)	0.7345 (0.0628)	0.7523 (0.0531)	0.7541 (0.0536)	0.7557 (0.0493)	0.7574 (0.0582)	0.7921 (0.0568)
Logistic regression	0.7343 (0.0575)	0.7847 (0.0312)	0.7981 (0.0356)	0.7984 (0.0371)	0.7951 (0.0529)	0.7986 (0.0440)	0.8211 (0.0447)
ANN	0.7362 (0.0625)	0.7770 (0.0423)	0.7970 (0.0326)	0.8023 (0.0308)	0.8136 (0.0435)	0.8141 (0.0436)	0.8206 (0.0526)
XGboost	0.7428 (0.0545)	0.7884 (0.0312)	0.8045 (0.0301)	0.8083 (0.0304)	0.8074 (0.0375)	0.8207 (0.0283)	0.8026 (0.0392)
Light GBM	0.5000 (0.0000)	0.7194 (0.0683)	0.7564 (0.0410)	0.7923 (0.0396)	0.8104 (0.0410)	0.8267 (0.0338)	0.8234 (0.0474)
SVM	0.7319 (0.0482)	0.7745 (0.0490)	0.7891 (0.0304)	0.7943 (0.0408)	0.7965 (0.0452)	0.8005 (0.0569)	0.8130 (0.0507)

**Table 18 ijerph-19-15171-t018:** Results of Wilcoxon signed-rank test between random forest and comparative models using 10% of the training dataset.

	*p*-Value	
	Liver Cancer Risk Association	Hepatitis B Risk Association
Decision tree	0.0000	0.0000
Logistic regression	0.0000	0.0000
ANN	0.0000	0.0000
XGboost	0.0000	0.0000
LightGBM	0.0000	0.0000
SVM	0.0000	0.0000

**Table 19 ijerph-19-15171-t019:** AUC values of multi-disease risk association models and comparative models.

	Liver Cancer	Hepatitis B
Random forest	0.7680 (0.0535)	0.5350 (0.0579)
Decision tree	0.7430 (0.0577)	0.5437 (0.0669)
Logistic regression	0.7518 (0.0580)	0.6019 (0.0632)
ANN	0.7106 (0.0625)	0.5480 (0.0591)
XGboost	0.7944 (0.0531)	0.5472 (0.0444)
LightGBM	0.8037 (0.0482)	0.5564 (0.0450)
SVM	0.6949 (0.0696)	0.5183 (0.0344)

## Data Availability

The data presented in this study is publicly available (https://knhanes.kdca.go.kr/knhanes/sub03/sub03_01.do, accessed on 1 February 2022).

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
