# Peer review of "Risk Association of Liver Cancer and Hepatitis B with Tree Ensemble and Lifestyle Features"

_ijerph, 2022, doi:10.3390/ijerph192215171_

Round 1
Reviewer 1 Report (Previous Reviewer 1)
The article "Predicting Risk of Liver Cancer and Hepatitis B with Tree
Ensemble and Lifestyle Features" by Eunji Koh and Younghoon Kim develop classical machine learning model for risk prediction for above mentioned diseases from lifestyle features. There overall scientific design of the article looks valid. The possibility to use methods like SVM on highly unbalanced or small dataset like used for cancer risk prediction require some additional comments. There is also a minor problem with the acknowledgments section of the article taken from the article template.
Author Response
Please see the attachment.

Reviewer 2 Report (New Reviewer)
This study proposed several risk prediction models for liver cancer and hepatitis B, and found that random forest model had the highest efficacy. Though the manuscript presented some significant findings, I still had some concerns on the study.
Major issues
1. The authors used KNHANES data to construct the model and the dataset covered seven years. However, we could not identify the study design, cohort or cross-sectional? If the authors used the cross-sectional data, the model was an association model, but not a risk predicting model.
2. In the method section, the details of KNHANES data were not provided. Is it representative to Korean people? If not, I doubt its generalizability. Since authors used lifestyle data to increase the efficiency, we need to know the details of the data source.
Minor issues
1. What approaches did the authors process to minimize the bias?
2.Though random forest had the highest efficacy, the difference with other models was significant or not. Authors needs to conduct ROC curve to compare the diagnostic efficacy between models.
Based on the above issues, authors need to revise the manuscript.
Round 2
Reviewer 2 Report (New Reviewer)
This study was designed as a cross-sectional mode therefore the outcome was an association not a prediction. Since in a cross-section study, the exposure and outcome occurrence did not have any time sequence, we could not conduct any prediction. Possibly, the lifestyle factors and liver disease happened at similar period and the disease affected the lifestyle. I do not think the risk predicting is of value, based on the current lifestyle features.
Random forest method was a key tool for the risk model, but in table 7 and 18 there were still some other models having the same efficacy. I could not see the advantages of random forest for the risk prediction.
I could not see the novelty or practical application of the prediction model and give any acceptance vote.
Author Response
Please see the attachment

This manuscript is a resubmission of an earlier submission. The following is a list of the peer review reports and author responses from that submission.
Round 1
Reviewer 1 Report
The article "Predicting Risk of Liver Cancer and Hepatitis B with Lifestyle
Features" by Eunji Koh and Younghoon Kim describes possibility to detect cancer risk basing only on the lifestyle features. The overall scientific hypothesis deserves consideration, but the presentation of results requires the major rewrite. Introduction section of the article shows a large confusion between cancer detection (diagnosis) basing on lifestyle features (seems not possible task now and in near future basing on my knowledge) and cancer risk prediction (that is solvable task). The introduction section also contain descriptions of many models based on diagnostic features that even more increasing confusion between diagnosis and risk estimation. The results part contain mostly technical results and discussion about training the various machine learning models, but not the results for accuracy of risk estimation and other value-based metrics of the proposed model. I suggest to move all the training details and metrics into the methods section of the article.
Reviewer 2 Report
In this manuscript the authors aimed to develop models for predicting the risk of liver cancer and hepatitis B with lifestyle features only rather than medical diagnosis results. The authors work is interesting and It shows the potentials of using lifestyle features to predict the risk of liver cancer and hepatitis B.
However, the manuscript is difficult to follow, it needs to be restructured, there is a lot of unnecessary repetition, and some missing information.
Abstract
- I don’t think you need to mention the source of the data this way, you can go with Korea National Health and Nutrition Examination Survey (KNHANES) 2018, and remove who did the survey. So it will be something like ”In this work, we use the publicly available data from Korea National Health and Nutrition Examination Survey (KNHANES) 2018 to develop a artificial neural network models for predicting the risk of liver cancer and hepatitis B.“
- I don’t think you should refer to your models as framework, since a framework would require more work on the pre-processing.
- You repeat the work aim twice, “We use neural networks …” and “We proposed a neural network…” that shouldn’t be the case.
- You should mention that you compared your models to other classical machine learning models.
- You should report very briefly the performance results of your models, before the last concluding sentence.
Introduction
- In General, you might need to check the language. Many sentences just don’t sound natural, which make difficult to understand your point.
- I think it will be better to start the introduction with talking about the liver cancer and hepatitis B. Similar to Lines 22-26. Then you can talk about what you mentioned in the first paragraph.
- Lines 39-45 should be before you start talking about others work. Maybe at the beginning after talking about the cancer as suggested.
- You don’t need to go very much in details about others work and results, just mention maybe that different approaches that used different machine learning models were used and that all of them in a way or another used clinical variables/features in there models.
- You really need a discussion section. There you can go more into details about the others work results and compare it to your work and your results.
- Line 48: it is “radiomics”.
- You use some abbreviations before defining them. Line 33, SVM used before it was defined.
- You also set abbreviations but then don’t use them later in the text when they appear like in Line 184 “Logistic regression [36], random forest [37], and decision tree”.
- Lines 72-87 just too long. I don’t think you need to put them in points, you can say something like “in this work we developed and evaluated artificial neural network models for predicting the risk of liver cancer and hepatitis B using lifestyle features only.” The remaining description and elaboration on this can go to the discussion section, after you make it.
Data and Methods
- In General, there is a lot of repetition, unnecessary sentences, assumptions that shouldn’t be in this section.
- You should put a reference to the KNHANES.
- The assumptions and the excluded features shouldn’t be mentioned here, maybe you can talk about it in the discussion.
- The number of datasets you used should ne mentioned before section 2.2, at the beginning of 2.1, somewhere in the first paragraph briefly.
- In section 2.2, you don’t need to explain the difference between the statistical tests. Just t-test for continuous variables, and chi for categorical.
- Be clear that the tests were between cancer vs no cancer, HBV vs no HBV.
- In 2.3, I don’t think you need to describe what are the imputation approaches, since you have references to them. Same for the neural networks hyperparameters, You don’t need to explain what is hidden layer or adam or solver, etc. You can cut a lot here. You mainly want to mention that you tried to optimized the following hyperparameters and the optimized parameters reported in table 4.
- In 2.4, don’t go and describe what are the LR, DT, etc… also you need to mention the hyper parameters you used for them when you trained them for the comparison.
- The comparison between ANN and LR,DT,..etc should be in the discussion section when you describe why maybe your model performed better than them.
Results
- In General, a lot of repetition, many sentences should be moved to a discussion section to highlight the findings and their meaning and importance.
- Figures 2,3 and 4, need legend to explain the difference between the curves of different colours.
- You might need to add one or two more decimals to the AUC results,
- since they are close to each other. E.g., 0.97 à9701.
- You need to perform significancy test to see if there are significant difference between the compared models, the training/testing models, and the different models with different imputations approaches.
- Section 3.4 should be separated, and the discussion should be a major section by itself, not under results.
Conclusion
- The limitations put them in a previous discussion section and elaborate on them.
Round 2
Reviewer 1 Report
The article is substantially improved, but I would suggest another round of checks done either by editor, or during another round of reviews. The all links in article PDF I've got are broken and it is hard to follow the references this way. As a development of the article topic I would suggest to use DSM (design structure matrix) optimization approach of interlinks between the cancer and Hepatitis B to visualize the features and their combinations into the highly connected sets. In the current variant article looks good enough to be published if the problem with links will be fixed and some minor final spellchecks will be done.
Reviewer 2 Report
The authors have made changes and addressed the comments I sent before. However, there still some comments.
In General, the figures, tables, pages and references numbers are missing, I only see a question mark instead. Please fix that.
Abstract
- In line 4, you need couple of words about the importance of the prediction models, something like “Early prediction of liver cancer can help preventing fatal conditions. Therefore, in this study we propose a risk prediction…”
- In line 5, maybe you can mention the size of the dataset, something like “A dataset of 3300 subject from the Korea National Health and Nutrition Examination Survey (KNHANES) 2018 was used”
- In line 8-10, maybe you should define the number of the compared models “The performance of our proposed method was compared with three classical machine learning models. The results showed that the proposed method outperformed the other models with an area under receiver operator characteristic curve of 0.974.”
- There is no need for the abbreviations since you don’t use them in your abstract. But you should define these (ANN, AUC) the first time they mentioned in the main body (not abstract).
Introduction
- The first paragraph needs references.
- Line 28, that is a strong claim, machine learning can’t always find significant relationship. Maybe you can say something like “The machine learning methods have the potentials to find significant relationships …”.
- Line 50, maybe add” in this study we propose …”.
Data and Methods
- Line 99, set a reference for GridSearchCV.
- discussion section when you describe why maybe your model performed better than them.
Results
- Table 7 results might need to perform a test for correction for multiple comparisons.
Conclusion
- I think you should consider moving some of the sensitivity and other results to the results section, but discuss the meaning of the findings in the discussion.
- You also need to discuss more why your method is better than the random forest in figure 3. It is very close results. Elaborate on why then you model is better.
- You might need to consider starting the discussion with a different paragraph and improve the limitations paragraph.
- In general, the discussion section need some organizing, references for the claims and comparison to other people work.
Conclusion
- The limitations put them in a previous discussion section and elaborate on them a bit more. Remove them from the conclusion,
